# Safe Opponent-Exploitation Subgame Refinement

## Abstract

Search algorithms have been playing a vital role in the success of superhuman AI in both perfect information and imperfect information games. Specifically, search algorithms can generate a refinement of Nash equilibrium (NE) approximation in games such as Texas hold'em with theoretical guarantees. However, when confronted with opponents of limited rationality, an NE strategy tends to be overly conservative, because it prefers to achieve its low exploitability rather than actively exploiting the weakness of opponents. In this paper, we investigate the dilemma of safety and opponent exploitation. We present a new real-time search framework that smoothly interpolates between the two extremes of strategy search, hence unifying safe search and opponent exploitation. We provide our new strategy with a theoretically upper-bounded exploitability and lower-bounded reward against an opponent. Our algorithm enables computationally efficient online adaptations to a possibly changing opponent model. Empirical results show that our method significantly outperforms NE baselines when opponents play non-NE strategies and keeps low exploitability at the same time. It is also much more efficient than previous safe exploitation baselines.

## 1 Introduction

Behind the recent breakthroughs of superhuman AIs in Go (Silver et al., 2016; 2017; Schrittwieser et al., 2020), heads-up no-limit Texas hold'em (HUNL) (Brown et al., 2018; Moravcík et al., 2017; Brown & Sandholm, 2019; Brown et al., 2020), and Hanabi (Lerer et al., 2020), search plays a vital role. In perfect information games, Monte Carlo tree search (MCTS) is widely applied to improve policy's strength. In zero-sum imperfect information games such as poker, search algorithms are used to find a Nash equilibrium (NE) approximation in subgames encountered in real time (Brown & Sandholm, 2017; Burch et al., 2014a). They are both theoretically sounded and empirically powerful. In fully-cooperative imperfect information games, the search algorithm proposed in Tian et al. (2020b) is proved to never be detrimental to the current policy. Lerer et al. (2020) also ensures original performance asymptotically.

In zero-sum games, NE-based search algorithms (Burch et al., 2014a; Moravcik et al., 2016; Brown & Sandholm, 2017; Brown et al., 2018) find safe strategies with low exploitability and produce strong baselines against all opponents (Brown & Sandholm, 2019). However, it may be overly conservative confronted with opponents with limited rationality and fail to take advantage of their weaknesses to obtain higher rewards (McCracken & Bowling, 2004; Johanson et al., 2007; Li & Miikkulainen, 2018). From the other perspective, there have been extensive studies on opponent exploitation to address the problem. Some typical works (Carmel & Markovitch, 1996; Billings et al., 2003; Gilpin & Sandholm, 2006; Li & Miikkulainen, 2018) model the opponent's strategy based on previous observations and then search for a new strategy to exploit this model. However, these methods often neglect the significance of the strategy safety, thus being highly exploitable by the opponent. Few exceptions including Johanson et al. (2007) and Ganzfried & Sandholm (2015a) aimed to search for safe and robust counter strategies, but they are computationally inefficient in an online setting where the opponent model is being updated continuously with streamed data.

In this paper, we study the dilemma of **safety** and **opponent exploitation** and present a new scalable real-time search framework **Safe Exploitation Search (SES)** that smoothly interpolates between the two extremes of strategy search, hence unifying safe search and opponent exploitation. It enables

computationally efficient online adaptations to a continuously changing opponent model, which is hard to address by previous safe exploitation algorithms. The safety criterion requires the refined strategy to stay close to NE, formally speaking, to expose limited exploitability against any opponents, while the opponent exploitation criterion requires the strategy to adapt to its specific opponent and to exploit its weaknesses. We propose a novel maximization objective which combines the safety objective and exploitation, controlled by the exploitation level $\alpha$. We construct a new gadget game to optimize this objective, which enables our method's scalability to large games such as Texas Hold'em. Theoretically, we prove that SES is guaranteed to outperform NE at the cost of some constant increase in its own exploitability confronted with non-NE opponents. Empirically, we evaluate the effectiveness of our search algorithm in 1 didactic matrix game 2 poker games: *Leduc Hold'em* (Southey et al., 2005) and *Flop Hold'em Poker* (FHP)(Brown et al., 2019). The experiment results demonstrate that our algorithm significantly outperforms NE baselines against non-NE opponents and keeps low exploitability at the same time. Additionally, SES is much more computationally efficient than previous safe exploitation baselines.

## 2 RELATED WORK

This paper investigates the problem of safe opponent exploitation in two-player zero-sum imperfect information games. We propose a novel search algorithm which balances between NE and exploiting opponents. Two major relevant research areas are search algorithms in imperfect information games, and opponent exploitation.

**Search in imperfect information games.** In recent literature, search techniques are witnessed to be important in developing strong AI strategies in both perfect and imperfect information games (Burch et al., 2014a; Moravcik et al., 2016; Brown & Sandholm, 2017). Texas hold 'em poker is widely employed as a benchmark for imperfect information games. A primary part of the long-term research on Texas hold'em poker is the evolution of subgame solving algorithms, which aim at achieving a more accurate Nash equilibrium approximation in the subgame encountered given a pre-computed strategy for the full game which we refer to as the blueprint strategy. Unsafe search (Billings et al., 2003; Ganzfried & Sandholm, 2015b; Gilpin & Sandholm, 2006; 2007) estimates the subgame reach probability assuming the opponent follows blueprint, and searches for a refined subgame strategy. Subgame resolving (Burch et al., 2014a) and maxmargin search (Moravcik et al., 2016) are theoretically sounded **safe** search algorithms which ensure that the subgame strategy obtained is no worse than the blueprint. They search in a gadget game and achieve safety by providing the opponent with the option not entering the current subgame. DeepStack (Moravcík et al., 2017) and Libratus (Brown et al., 2018) build strong poker AIs with the aid of search. Beyond poker, search algorithms for subgame refinement have also shown promise in improving joint strategies in cooperative imperfect information games such as Hanabi (Lerer et al., 2020) and the bidding phase of contract bridge (Tian et al., 2020a). The purpose of our search algorithm is different from previous methods in poker literature. We seek to exploit opponents while keeping exploitability low, rather than simply approximating NE.

**Opponent exploitation.** Most previous opponent exploitation researches (Carmel & Markovitch, 1996; Billings et al., 2003; Gilpin & Sandholm, 2006; Li & Miikkulainen, 2018) typically model the opponent's strategy based on previous observations and then search for a new strategy to exploit this model, but put little emphasis on safety.

One similar work is Johanson et al. (2007) which proposes $p$-restricted Nash response (RNR) to find a safe exploitation strategy to the estimated opponent's strategy. It calculates a Nash equilibrium for the whole game restricting that the opponent plays the estimated strategy $\sigma^{\text{fix}}$ with probability $p$, and any strategy with probability $1 - p$. In that paper, Johanson et al. (2007) prove that a $p$-RNR to $\sigma^{\text{fix}}$ is Pareto optimal with respect to exploitation and safety. However, it does not provide an explicit bound. Additionally, whenever the estimated opponent model changes or we want to use a different $p$ to balance between safety and exploitation, $p$-RNR has to recompute the strategy for the whole game. It is computationally inefficient in an online setting, where the opponent model is updated after every round with new game data. Our algorithm instead takes modelling error into account and provides explicit bounds for both safety and exploitation. With the aid of real-time search, it only searches for strategies in subgames encountered instead of the whole game. Our experiments show that it is more efficient than Johanson et al. (2007).

Ganzfried & Sandholm (2015a) study safe exploitation strategies in repeated games, which is a different setting from this paper. Intuitively, it achieves safety by risking in exploitability at most what it has earned over NE in expectation in previous rounds. Therefore, its expected value in the whole repeated game is never worse than the NE. In contrast, this paper focuses on the safety of stage game strategies. Furthermore, our algorithm is complementary to Ganzfried & Sandholm (2015a). Ganzfried & Sandholm (2015a) calculate an $\varepsilon$-safe best response for the whole game at each iteration with LP. This procedure is one of the main limitations on the algorithm's scalability. Our algorithm, which only refines strategies in subgames in real-time, can be a possible substitute for LP.

To our knowledge, we are the first paper to investigate the safe opponent exploitation problem in subgame resolving schemes. Subgame resolving enables online adaptations to a continuously changing opponent model, eliminating the need to recompute a whole game strategy. It offers computational benefits in practical opponent exploitation circumstances.

It is worthwhile mentioning that there also has been extensive research (Albrecht & Stone, 2018) on agent modeling. However, this paper only focuses on the theoretical and empirical results of the search algorithm, but not the agent modeling techniques. We can use off-the-shelf agent modeling algorithms to estimate opponent's strategies. Agent modeling provides the ability to reason about the others, making predictions of their strategies, types, etc. He et al. (2016) embeds opponent model learning into deep Q-learning, which stabilizes the training process faced with opponents whose policy is changing over time. LOLA (Foerster et al., 2018) considers the evolution of other agents in the training process of multi-agent reinforcement learning. Raileanu et al. (2018) proposes to model others from one's own policy. Rabinowitz et al. (2018) uses meta learning to predict strategies of different kinds of agents.

## 3 NOTATIONS AND BACKGROUND

An extensive-form imperfect information game $G = (P, H, Z, A, \chi, \rho, \cdot, \sigma_c, u, \mathcal{I})$ describes sequential interactions among agents, where agents have private information. A finite set $P$ consists of $n$ players and a chance node $c$ which represents the stochastic nature of the environment. The set of non-terminal decision nodes is denoted as $H$, and $Z$ is a set of terminal nodes, or leaves. The set of possible actions is $A$, and $\chi : H \to 2^{|A|}$ is a function which assigns to each decision node $h \in H$ a set of legal actions. A player function $\rho : H \to P$ assigns to each decision node a player $p \in P$ who acts at that node. If action $a$ leads from $h$ to $h'$, we write $h \cdot a = h'$. If there exists a sequence of actions leading from $h$ to $h'$, we write $h \sqsubseteq h'$. At each node $h \in H$, the acting player $p = \rho(h)$ picks an action from legal actions $a \in \chi(h)$, and leads node $h$ into its child $h \cdot a$. The chance node always samples an action from its own distribution $\sigma_c$, which is common knowledge to all players. Utility functions are $u = (u_1, u_2, \ldots, u_n)$, where $u_i : Z \to \mathbb{R}$ defines the utility of player $i$ at terminal node $z \in Z$. The nature of imperfect information is characterized by infosets $\mathcal{I} = (\mathcal{I}_1, \mathcal{I}_2, \ldots, \mathcal{I}_n)$, where $\mathcal{I}_i = (I_{i,1}, \ldots, I_{i,k_i})$ is a partition of $H$ for player $i$. Two states in the same infoset must have the same acting player and the same legal action sets. We use $I(h)$ to denote the infoset that $h$ belongs to. A player $p$ cannot distinguish between states $h_1$ and $h_2$ if $I(h_1) = I(h_2)$, and thus should behave identically on all states in the same infoset.

The strategy of a player $p$ is $\sigma_p : \mathcal{I}_p \times A \to \mathbb{R}$, where $\sigma_p(I, a)$ is a distribution over valid actions on infoset $I$. For simplicity, we also use $\sigma_p(h, a)$ to denote $\sigma_p(I(h), a)$. We use $\pi^\sigma(h)$ to denote the probability of reaching state $h$ from the root when agents choose a strategy profile $\sigma = \langle \sigma_1, \sigma_2, \ldots, \sigma_n \rangle$. Formally, $\pi^\sigma(h) = \prod_{h' \cdot a \sqsubseteq h} \sigma_{\rho(h')}(h', a)$. We use $\pi^\sigma_{-p}(h) = \prod_{h' \cdot a \sqsubseteq h \land \rho(h') \neq p} \sigma_{\rho(h')}(h', a)$ to denote the probability of reaching $h$ when player $p$ always chooses the action that leads to $h$ whenever possible. $\pi^\sigma(h, h')$ is the reaching probability of $h'$ from $h$. $\pi^\sigma(h \cdot a, h')$ is the the probability of reaching $h'$ from $h$ if action $a$ is taken at $h$. These probabilities can be formally defined in a similar manner.

The expected utility of player $p$ given strategy profile $\sigma$ is $u^\sigma_p = \sum_{z \in Z} \pi^\sigma(z) u_p(z)$. The **counterfactual value** $v^\sigma_p(I, a)$ is the expected utility that player $p$ will obtain after taking action $a$ at infoset $I$, given the joint policy profile is $\sigma$. Mathematically, it is the weighted sum of expected values at all states $h \in I$.

$$v^\sigma_p(I, a) = \frac{\sum_{h \in I, z \in Z} \pi^\sigma_{-p}(h) \pi^\sigma(h \cdot a, z) u_p(z)}{\sum_{h \in I} \pi^\sigma_{-p}(h)} \tag{1}$$

In the rest of the paper, we focus on two-player zero-sum games with perfect recall. Zero-sum means $\forall z \in Z, u_1(z) + u_2(z) = 0$. Perfect recall means that no player will forget the information which has been obtained previously in the game. This is a common assumption in related literature.

A best response strategy $BR_p(\sigma_{-p}) = \arg\max_{\sigma_p} u_p^{\langle \sigma_p, \sigma_{-p} \rangle}$ for player $p$ is the strategy that maximize his own expected utility against fixed opponent strategy $\sigma_{-p}$. The **exploitability** of strategy $\sigma_p$ is $\exp(\sigma_p) = u_p^{\sigma^*} - u_p^{\langle \sigma_p, BR_{-p}(\sigma_p) \rangle}$ where $\sigma^*$ is the optimal strategy, and is an NE in two-player zero-sum games. It measures the performance of $\sigma_p$ against its best response comparing with the NE. A **counterfactual best response** $CBR_p(\sigma_{-p})$ is a strategy where $\sigma_p(I, a) > 0$ if and only if $v_p^\sigma(I, a) \geq \max_b v_p^\sigma(I, b)$. Counterfactual best response is a best response, but not vice versa. The **counterfactual best response value** $CBV_p^{\sigma_{-p}}(I) = v_p^{\langle CBR_p(\sigma_{-p}), \sigma_{-p} \rangle}$ is the expected utility of the counterfactual best response policy. Since we focus on two-player zero-sum games, we will use $CBV^{\sigma_p}(I)$ as a shorthand notation for $CBV_{-p}^{\sigma_p}(I)$.

We follow the imperfect information subgame definition as in Burch et al. (2014b). An **augmented infoset** contains states which cannot be distinguished by the remaining players.

**Definition 1.** *An imperfect information subgame $S$ is a forest of trees, closed under both the descendant relation and membership within augmented infosets for any player. Let $S_{\text{top}}$ be the set of nodes which are roots of each tree in $S$.*

## 4 METHOD

In this section, we introduce our novel search algorithm called safe exploitation search (SES), which exploits the weaknesses of opponent while ensuring a bounded exploitability. Let $\sigma$ be the pre-computed blueprint strategy. Without loss of generality, assume we search for player 2's refined strategy $\sigma_2^S$ by applying SES to all subgames $S \in \mathbb{S}$. Finally, the refined strategy for P2 after search is $\sigma_2'$, which is the same as $\sigma_2$ in $\{I_2^i | \forall S \in \mathbb{S}, I_2^i \notin S\}$ and is replaced with $\sigma_2^S$ in $S \in \mathbb{S}$.

### 4.1 SAFE EXPLOITATION SEARCH

Our algorithm offers a unified approach to balance between these two demands with theoretical guarantees. The objective of our search algorithm is to find a new subgame strategy $\sigma_2^S$ for $S \in \mathbb{S}$ which maximizes

$$SE(\sigma_2^S) = \alpha \sum_{I_1^j \in S_{\text{top}}} \hat{p}(I_1^j) \left( v_1^\sigma(I_1^j) - CBV_1^{\sigma_2^S}(I_1^j) \right) + (1 - \alpha) \min_{I_1^j \in S_{\text{top}}} \left( v_1^\sigma(I_1^j) - CBV_1^{\sigma_2^S}(I_1^j) \right),$$
(2)

where $\alpha \in [0, 1]$ is a hyper-parameter controlling the exploitation level, and $\hat{p}(I_1^i)$ is the estimated probability of player 1 entering infoset $I_1^j \in S_{\text{top}}$. Given P2's strategy (which is the blueprint $\sigma_2$) and P1's actual strategy (which does not have to be the blueprint $\sigma_1$), the real probability of player 1 entering infoset $I_1^j \in S_{\text{top}}$ (which we denote as $p(I_1^j)$) is determined. $\hat{p}(I_1^i)$ is just an estimation of $p(I_1^j)$. For instance, in poker, it is the estimated distribution of private cards player 1 holds. Both theoretically and empirically, such estimation does not have to be fully accurate. It can be done with off-the-shelf opponent modeling techniques, which lies beyond the focus of this paper.

Though seemingly complicated, intuitively, the maximization objective achieves a balance between **opponent exploitation** and **safety**, controlled by **exploitation level** $\alpha$. The first part of the objective is maximized when $\sum_{I_1^j \in S_{\text{top}}} \hat{p}(I_1^j) CBV_1^{\sigma_2^S}(I_1^j)$ is minimized. It aims at finding a strategy $\sigma_2^S$ which results in the lowest value for P1 under the assumption that the reach probabilities is $\hat{p}$. It can be interpreted as exploiting the estimated P1's strategy. The second part of the objective demands the resolved strategy to behave well against any reach probability distribution. We use the subgame margin $\min_{I_1^j \in S_{\text{top}}} \left( v_1^\sigma(I_1^j) - CBV_1^{\sigma_2^S}(I_1^j) \right)$ (Moravcik et al., 2016) which can be regarded as the worst-case utility increase for P2.

By maximizing the objective 2, we provide sound theoretical results for both safety and opponent exploitation. Additionally, we provide analyses of how (1) exploitation level $\alpha$, (2) accuracy of

opponent modeling, and (3) strength of the blueprint strategy impact the theoretical bound. By gradually increasing $\alpha$ from 0 to 1, our algorithm tends to exploit rather than keeping safety.

**Theorem 1.** (safety) *Let $\mathbb{S}$ be a disjoint set of subgames $S$. Let $\sigma^* = \langle \sigma_1^*, \sigma_2^* \rangle$ be the NE where P2's strategy is constrained to be the same with $\sigma_2$ outside $\mathbb{S}$. Define $\Delta = \max_{S \in \mathbb{S}, I_1^i \in S_{top}} |CBV_1^{\sigma_2^*}(I_1^i) - v_1^\sigma(I_1^i)|$. Let $\tilde{p}(I_1^i)$ be the reach probability given by $\sigma_1^*$. Let $\hat{p}(I_1^i)$ be the estimation of reach probability $p(I_1^i)$ given by the real opponent strategy. Define $\tau = \max_{S \in \mathbb{S}, I_1^i \in S_{top}} |\frac{\hat{p}(I_1^i) - \tilde{p}(I_1^i)}{\tilde{p}(I_1^i)}|$. Whenever $1 - (2\tau + 1)\alpha > 0$, we have a bounded exploitability given by:*

$$\exp(\sigma_2') \le \exp(\sigma_2^*) + \frac{2}{1 - (2\tau + 1)\alpha}\Delta. \tag{3}$$

Recall that $\sigma_2'$ is the refined strategy after search. The proof is provided in Appendix A. This theorem implies that the exploitability of the new strategy is smaller than that of strategy $\sigma_2^*$ plus a constant value, which is the closest strategy to NE if constrained to differ from $\sigma_2$ only in $\mathbb{S}$. The corresponding theoretical result of maxmargin search (Moravcik et al., 2016), a safe search algorithm with no opponent exploitation abilities, is $\exp(\sigma_2') \le \exp(\sigma_2^*) + 2\Delta$. Comparing these two results, we can interpret the term $2/(1 - (2\tau + 1)\alpha)$ as the additional risk introduced by exploiting the opponent. If exploitation level $\alpha = 0$, then our bound is as tight as that of maxmargin search (Moravcik et al., 2016). The bound also gets tighter if the $\tau$ gets smaller, or the blueprint $\sigma_2$ is closer to $\sigma_2^*$.

**Theorem 2.** (opponent exploitation) *Let $\epsilon = \|\hat{p} - p\|_1$ be the L1 distance of the distribution $p(I_1^i)$ and $\hat{p}(I_1^i)$. Let $\eta = \min_{S \in \mathbb{S}} \max_{I_1^j \in S_{\text{top}}} \left( CBV_1(I_1^j, \sigma_2^S) - CBV_1(I_1^j, \sigma_2^*) \right) \ge 0$. We use $BR_p^{[\mathbb{S}, \sigma_p]}(\sigma)$ to denote the strategy for player $p$ which maximizes its utility in subgame $S \in \mathbb{S}$ against $\sigma_{-p}$ under the constraint that $BR_p^{[\mathbb{S}, \sigma_p]}(\sigma)$ and $\sigma_p$ differs only inside $\mathbb{S}$. By maximizing objective 2, for all $S \in \mathbb{S}$, the refined strategy $\sigma_2'$ satisfies*

$$u_2^{\left\langle BR_1^{[\mathbb{S}, \sigma_1]}(\sigma_2'), \sigma_2' \right\rangle}(S) \ge u_2^{\left\langle BR_1^{[\mathbb{S}, \sigma_1]}(\sigma_2^*), \sigma_2^* \right\rangle}(S) + \frac{1 - \alpha}{\alpha}(\eta - 2\Delta) - \epsilon\eta \tag{4}$$

The proof is provided in Appendix A. Observe that the reach probability $p$ is characterized by P1's strategy outside $\mathbb{S}$ and $\hat{p}$ is its estimation. Because the search algorithm always find a stronger response strategy for P1 in $\mathbb{S}$ (which is exactly $BR_1^{[\mathbb{S}, \sigma_1]}(\sigma_2')$) as well, opponent exploitation refers to adapting to P1's strategy $\sigma_1$ outside $\mathbb{S}$. This theorem implies that the utility of the new strategy $\sigma_2'$ is lower bounded by the utility of $\sigma_2^*$ when both confronted with P1's unknown strategy outside $\mathbb{S}$. It provides theoretical guarantees for the opponent exploitation ability of our algorithm. $\epsilon$ can be interpreted as estimation error. The lower bound increases if the estimation error get smaller or the blueprint $\sigma_2$ is closer to $\sigma_2^*$. We show empirically how exploitation level $\alpha$ and estimation error impact both safety and exploitation abilities in section 5.

## 4.2 GADGET GAME

In order to find $\sigma_2^S$ which maximize objective 2, a straight-forward method is to reformulate the maximization problem as a Linear Programming problem (Moravcik et al., 2016). However, LP solvers (Koller et al., 1994) cannot handle large-scale problems. Alternatively, inspired by Moravcik et al. (2016), we create a gadget game and then apply iteration-based NE algorithms such as CFR (Zinkevich et al., 2007; Tammelin et al., 2015; Lanctot et al., 2009) in the gadget game. The gadget game is carefully designed such that the NE solution found in it is exactly the solution to the original optimization problem.

As shown in Figure 1, the original subgame is copied into two identical parts $S_1, S_2$ in the gadget game. Player 2's infosets stretch over both branches, while player 1 can distinguish between the two parts. The procedure of constructing such gadget game can be summarized into 4 steps: (i). Create a chance node at the top of the gadget game. (ii) For the left part of the gadget game, we construct a P1 node to let P1 choose an infoset $I_1^i$ to enter. The following chance node samples a specific state with probability proportional to $\pi_{-1}^\sigma(h)$ for all $h \in I_1$. (iii) For the right part, create a chance node sampling an infoset $I_1^i$. The following chance node again samples a specific state with probability

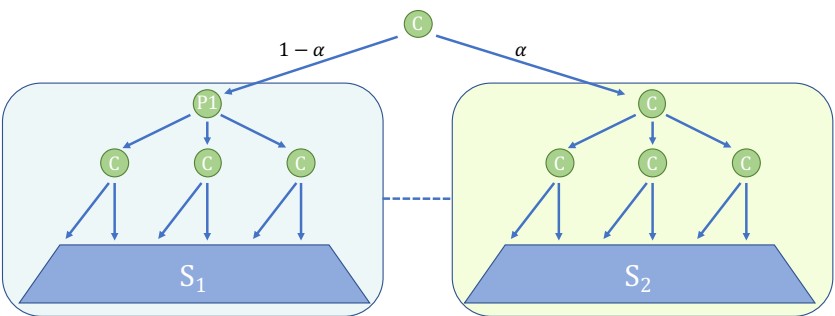

Figure 1: The gadget game of SES. The shadow and dashed line indicate that player 2 cannot distinguish between the two branches. $C$ represents chance node, $P1$ represents player 1's action node. $S_1$ and $S_2$ are two identical copies of the subgame $S$ with utility shifted.

| P1 \ P2 | L | M | R |
|---|---|---|---|
| U | 3 | 2 | 4 |
| O | 2 | 3 | 9.9 |
| D | 3 | 2 | 9.9 |
| F | -100 | -100 | 10 |

Table 1: The payoff matrix of the example zero-sum matrix game. The values are the payoffs of player 2. We will resolve for player 2.

proportional to $\pi_{-1}^{\sigma}(h)$ for all $h \in I_1$. (iv). Shift the utility of the gadget game by $v_1^{\sigma}(I_1^i)$. The details are described below.

1. The chance node at the top goes to the left part with probability $1 - \alpha$, and the right part with probability $\alpha$. The outcome is visible to P1 but not P2. Therefore, corresponding nodes in both branches are in the same infosets for P2, and his strategy $\sigma_2^S$ will be the same for both parts. Since $\sigma_1^S$ is the best-response to $\sigma_2^S$ and the two parts only differ at how to go to each infoset of player 1, player 1 will also keep his strategy the same in both parts.

2. We subtract $u_1(z)$ by $v_1^{\sigma}(I_1^i)$ for all $z \sqsubseteq h, h \in I_1^i$, and add $u_2(z)$ by $v_1^{\sigma}(I_1^i)$ in order to keep the subgame zero-sum. By doing so, the objective of a Nash Equilibrium of $p_2$ will change from maximizing $-CBV_1^{\sigma_2^S}(I_1^i)$ to maximizing $v_1^{\sigma}(I_1^i) - CBV_1^{\sigma_2^S}(I_1^i)$.

3. As for the left part of the gadget game, the P1 node on the second level in Figure 1 enables P1 to enter an arbitrary infoset $I_1^i$. Since this is a zero-sum game, in an NE strategy, he will enter the one with lowest $v_1^{\sigma}(I_1^i) - CBV_1(I_1^i, \sigma_2^*)$ which is exactly the minimization in the second term of $SE(\sigma_2^S)$.

4. The chance node on the second level of the right part will sample an infoset $I_1^i$ according to reach probability $\hat{p}(I_1^i)$. So that the NE objective of this part is exactly the summation in the first term of $SE(\sigma_2^S)$.

### 4.3 MATRIX GAME

In this part, we offer a matrix game as an example to show the necessity of considering safety and expected payoff simultaneously, and to demonstrate the superiority of SES over a simple mixing strategy, which follows a best response to the estimated opponent model with probability $\alpha$ and follows the blueprint with probability $1 - \alpha$.

In the matrix game shown in Table 1, let's consider two specific NEs. In both NEs, P1 will play L/M with $0.5$ probability each. P2 will play U/O with $0.5$ probability in the first NE and O/D with $0.5$ probability in the second NE. Suppose the blueprint strategy is the first NE. Consider the case when P1 plays a rather weak strategy that he will only play action R. We apply SES to search for P2's refined strategy.

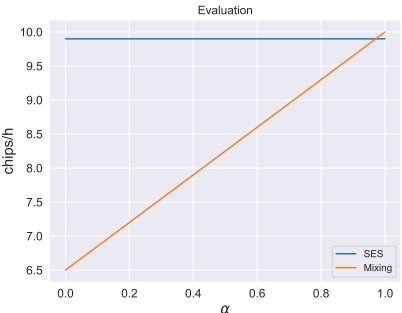 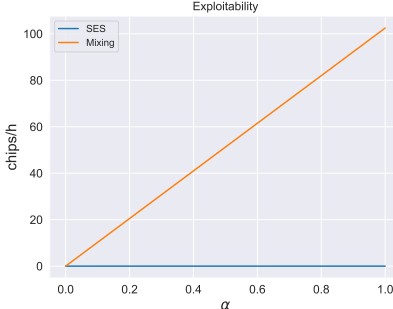

Figure 2: **Left:** Expected payoffs of SES and the mixing strategy in the proposed matrix game example. **Right:** Exploitability of the two algorithms.

When the estimation of opponent strategy is accurate such that $\hat{p} = p$, the best response of P2 is always playing F, which is highly exploitable, while SES finds the second NE under proper $\alpha$. To give more details, the exploitability and expected payoff of the strategy found by SES and the mixing strategy are shown in Figure 2. We can see that SES achieves both lower exploitability and better performance than the mixing strategy at almost all $\alpha$ values. The reason behind this success is that P1 always playing R is a Gift Strategy (Ganzfried & Sandholm, 2015a) in the designed matrix game and SES manages to utilize such gift strategy. Strategy $\sigma_{-p}$ is a Gift Strategy if it is not a best response to a NE strategy $\sigma_p^*$. Therefore, we can switch to $\sigma_p^*$ to get better performance against $\sigma_{-p}$ while also keeping exploitability low. However, the simple mixing strategy cannot find such "good" NE strategy so that it will perform much worse in both exploitability and expected payoff.

## 5 EXPERIMENT

Our experiment is done in *Leduc Hold'em* (Southey et al., 2005) and *Flop Hold'em Poker* (FHP) (Brown et al., 2019). *Leduc Hold'em* is a smaller-scale poker games and FHP is a larger one. The rules of these two pokers are provided in Appendix B. We demonstrate the exploitability and evaluation performance of SES against opponents of various strengths. The exploitability measures a search algorithm's safety, while head-to-head evaluation measures the ability of opponent exploitation. We also illustrate how estimation accuracy of opponent's strategy and the exploitation level $\alpha$ impact the results. Please refer to Appendix C for implementation details.

### 5.1 OPPONENTS

In our experiments, we test the performance of our algorithm against opponents of various strengths. For both Leduc Poker and FHP, we create 3 types of opponents with 3 random seeds each. The first type of opponent is an approximation of NE in the full game, and is regarded as a strong opponent. It is computed in the same way as the blueprint strategy with different seed. For the second and third type of opponents, we enumerate every infoset in the blueprint strategy and shift the action distribution randomly with probability $\mathrm{Pr_{shuffle}} = 0.3$ or $0.7$. We multiply the probability of each action by a random variable from $\mathrm{Uniform}(0,1)$, and then re-normalize the probability distribution. The procedure is motivated by Brown et al. (2018), in which such method is applied to create a number of diverse but reasonably strong agents. Even when $\mathrm{Pr_{shuffle}} = 0.7$, the strategy keeps close to NE with average L1 distance of each infoset 0.132 comparing to 1.036 of a random strategy to NE. So they are regarded as opponents who are not fully rational but with competitive strength.

### 5.2 SAFE OPPONENT SEARCH

In Figure 3, we demonstrate the head-to-head evaluation performances and corresponding exploitability of the refined strategies found by SES against opponents of various strengths, under different exploitation level $\alpha$ and estimation errors of opponent's strategy. Different lines in each plot refers to corresponding estimation error $\epsilon$, which is the L1 distance of $\hat{p}$ and $p$. We evaluate our refined

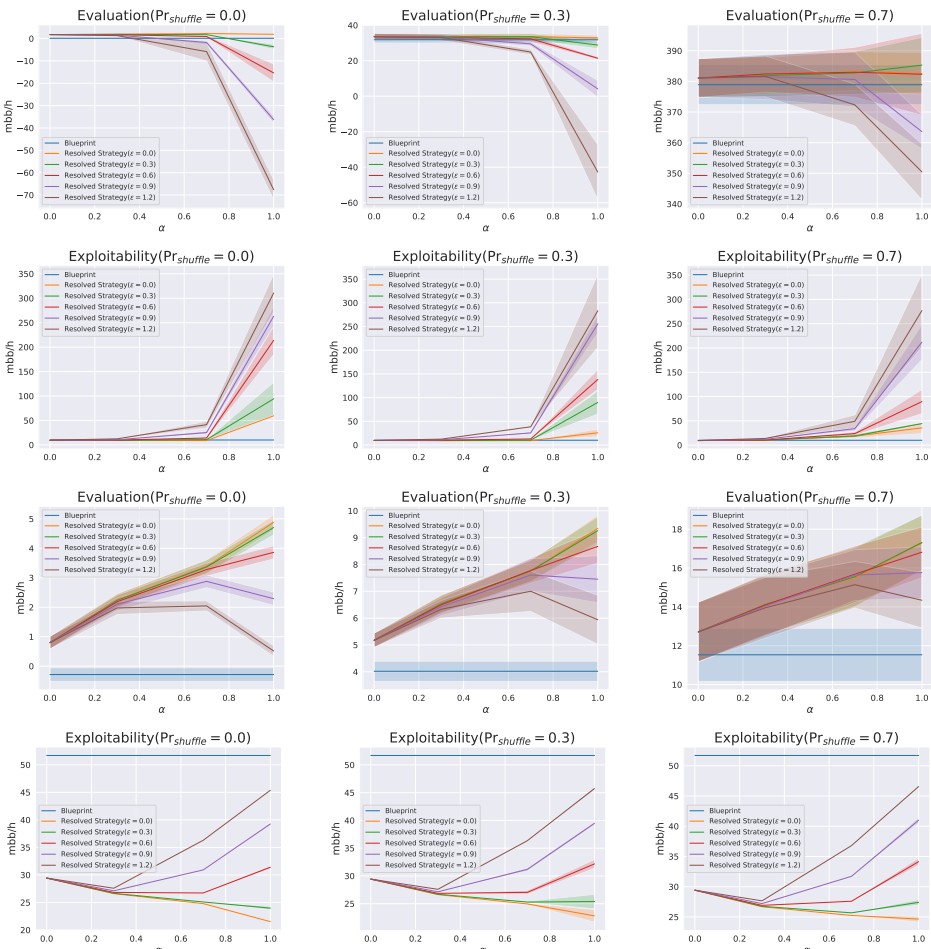

Figure 3: **Row 1&2:** Experiment results on Leduc poker. **Row 3&4:** Experiment results on FHP. From left to right, each column represents a type of opponent with $Pr_{shuffle} = 0.0, 0.3, 0.7$. The first row of each game shows the head-to-head expected payoffs against corresponding opponents, while the second row demonstrates the exploitability calculated for different refined strategies. The X-axis is the parameter $\alpha$ which controls exploitation level.

strategy when $\epsilon = 0.0, 0.3, 0.6, 0.9, 1.2$. The blue line is the result of blueprint strategy without conducting any search.

Generally speaking, SES balances between safety and opponent exploitation. The increase of exploitation level $\alpha$ helps win more chips from opponents, while resulting in the increase of the strategy's own exploitability. As can be seen in Figure 3, the exploitability increases when the exploitation level $\alpha$ grows from 0 to 1, which is consistent with Theorem 1. One exception is in FHP when $\epsilon$ is small: the exploitability surprisingly keeps decreasing even if SES puts more emphasis on opponent exploitation. Similar situations have also occurred in previous literature (Brown & Sandholm, 2017). The reason is that our opponent is quite close to NE outside the subgame which will make $\hat{p}$ close to $\tilde{p}$ when $\epsilon$ is small, which means the $\tau$ in Theorem 1 is small. As a result, we will have a low-exploitability resolved strategy when using unsafe search and the exploitability increases as $\epsilon$ increases.

When the estimation is completely correct ($\epsilon = 0.0$, the yellow line), the expected payoff in FHP increases as the exploitation level $\alpha$ grows higher. In Leduc poker, since the game is very small, the pre-computed blueprint is very close to NE. Therefore, when confronted with relatively strong opponents ($Pr_{shuffle} = 0.0, 0.3$) which are also close to NE, actually few things can be done other

than sticking with the blueprint. So the improvement introduced by SES is small. When facing relatively weak opponent ($\mathrm{Pr}_{\mathrm{shuffle}} = 0.7$), the improvement margin is slightly larger.

SES relies on an estimation of opponent's strategy. In order to test the robustness of our algorithm when the prediction of $p(I_1^i)$ is not accurate, we evaluate the performance of our algorithm with different values of estimation error $\epsilon$. As illustrated in Figure.3, the exploitability increases and the expected payoff drops when $\epsilon$ grows larger. The result is expected since an accurate estimation always provides benefits. However, it also demonstrates that SES can still achieve a trade-off between safety and opponent exploitation even when $\epsilon$ is considerably high. For instance, in FHP, $\epsilon$ is between 0 and 2, and $\epsilon = 1.2$ means that the predicted distribution is almost random. When $\epsilon \leq 0.6$, the expected payoff still keeps increasing with respect to $\alpha$. In case of a bad estimation, we can always choose smaller $\alpha$ to ensure safety.

### 5.3 COMPARISON WITH RESTRICTED NASH RESPONSE

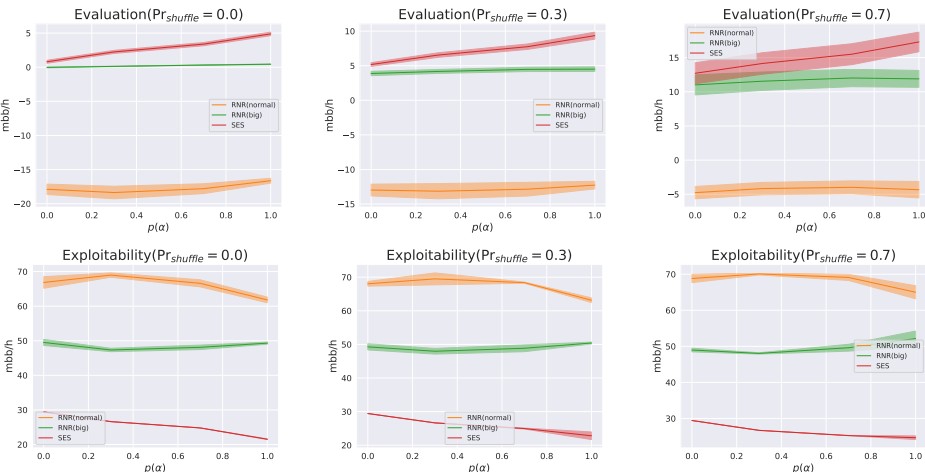

Figure 4: Comparison between SES and RNR. From left to right, each column represents a type of opponent with $\mathrm{Pr}_{\mathrm{shuffle}} = 0.0,\ 0.3,\ 0.7$. The X-axis is the parameter $\alpha$ for SES and $p$ for RNR.

We also compare SES with restricted Nash response (RNR) (Johanson et al., 2007) in FHP. In each round, we limit the computation time of RNR(normal) to 10 CPU second[1], which is the same for SES. However, as stated in section 2, RNR needs to recompute a strategy for the whole game in each round. It cannot converge in 10s. So we also compare with RNR(big), which has a budget of 10M CFR iterations in each round (around 190 CPU second in time). In contrast, SES only uses 10M CFR iterations to calculate its blueprint once. As is shown in Figure 4, SES significantly outperforms RNR(normal) in both exploitability and evaluation. SES also achieves much lower exploitability than RNR(big) and comparable evaluation results with much less computation time.

## 6 CONCLUSION

We propose a novel safe exploitation search (SES) algorithm which unifies both safe search and opponent exploitation. With the aid of real-time search, SES can make online adaptations to a changing opponent model. We also prove safety and opponent exploitation guarantees of SES in Theorem 1 and Theorem 2. The experimental results in our designed matrix game confirm the existence of the refined strategy which is both safe and actively exploiting the opponent. In games of poker, our method outperforms NE baselines while keeping exploitability low. SES is also much more efficient than previous safe exploitation algorithms without search. The exploitation level $\alpha$ is now regarded as a hyperparameter in our algorithm. However, ideally, $\alpha$ should be learnt automatically from opponents, and should be adaptive to opponent's strategy change. We leave this for future work.

---

[1]We test it on Intel(R) Xeon(R) Platinum 8276L CPU @ 2.20GHz

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

# A    PROOFS

## A.1    PROOF OF THEOREM 1

**Theorem 1.** *(safety) Let $\mathbb{S}$ be a disjoint set of subgames $S$. Let $\sigma^* = \langle \sigma_1^*, \sigma_2^* \rangle$ be the NE where P2's strategy is constrained to be the same with $\sigma_2$ outside $\mathbb{S}$. Define $\Delta = \max_{S \in \mathbb{S}, I_1^i \in S_{top}} |CBV_1^{\sigma_2^*}(I_1^i) - v_1^\sigma(I_1^i)|$. Let $\tilde{p}(I_1^i)$ be the reach probability given by $\sigma_1^*$. Let $\hat{p}(I_1^i)$ be the estimation of reach probability $p(I_1^i)$ given by the real opponent strategy. Define $\tau = \max_{S \in \mathbb{S}, I_1^i \in S_{top}} |\frac{\hat{p}(I_1^i) - \tilde{p}(I_1^i)}{\tilde{p}(I_1^i)}|$. Whenever $1 - (2\tau + 1)\alpha > 0$, we have a bounded exploitability given by:*

$$\exp(\sigma_2') \leq \exp(\sigma_2^*) + \frac{2}{1 - (2\tau + 1)\alpha}\Delta. \tag{5}$$

**Proof:**

For simplicity, we will omit the subscript of $CBV_1^{\sigma_2^*}$ by default. In order to prove Theorem 1, we will use mathematical induction on the level of the infoset. The depth $L$ has the same definition as in Brown & Sandholm (2017), i.e.

- For all the infosets which are direct parents of the subgames, we define $L(I) = 0$.
- For the infosets that are not ancestors of the subgames, we define $L(I) = 0$.
- For any infosets that are ancestors of the subgames, we define
  $L(I) = \max_{I' \in succ(I)} L(I') + 1$. That is, it has a higher level than any of its successors.

BASE CASE OF INDUCTION

Firstly, we will prove that for any infoset with level 0, the inequality of theorem 1 holds. For convenience, we consider that theorem 1 in a specific subgame $S$.

We will prove the infoset at the top of the subgame first. Since $SE(\sigma_2^*) \geq (1 - \alpha)(-\Delta) + \alpha \sum_i \hat{p}(I_1^i)(-\Delta) = -\Delta$, we have

$$(1 - \alpha) \min_{I_1^j} \left( v_1^\sigma(I_1^j) - CBV^{\sigma_2^S}(I_1^j) \right) + \alpha \sum_i \hat{p}(I_1^i)(v_1^\sigma(I_1^i) - CBV^{\sigma_2^S}(I_1^i))$$

$$\begin{aligned} &= SE(\sigma_2^S) \\ &\geq SE(\sigma_2^*) \\ &\geq -\Delta \end{aligned} \tag{6}$$

since $\sigma_2^S = \arg\max_{\tilde{\sigma}_2} SE(\tilde{\sigma}_2)$.

Furthermore, we have

$$\sum_i \hat{p}(I_1^i)(v_1^\sigma(I_1^i) - CBV^{\sigma_2^S}(I_1^i))$$

$$= \sum_i \hat{p}(I_1^i)(v_1^\sigma(I_1^i) - CBV^{\sigma_2^*}(I_1^i)) + \sum_i \hat{p}(I_1^i)(CBV^{\sigma_2^*}(I_1^i) - CBV^{\sigma_2^S}(I_1^i))$$

$$\leq \Delta + \sum_i \hat{p}(I_1^i)(CBV^{\sigma_2^*}(I_1^i) - CBV^{\sigma_2^S}(I_1^i)) \tag{7}$$

$$= \Delta + \sum_i \tilde{p}(I_1^i)(CBV^{\sigma_2^*}(I_1^i) - CBV^{\sigma_2^S}(I_1^i))$$

$$+ \sum_i (\hat{p}(I_1^i) - \tilde{p}(I_1^i))(CBV^{\sigma_2^*}(I_1^i) - CBV^{\sigma_2^S}(I_1^i))$$

where the second term $\sum_i \tilde{p}(I_1^i)(CBV^{\sigma_2^*}(I_1^i) - CBV^{\sigma_2^S}(I_1^i))$ is no larger than 0 because $\sum_i \tilde{p}(I_1)CBV^{\sigma_2^*}(I_1^i)$ is exactly what $\sigma_2^*$ minimized. Otherwise, $\sigma_2^*$ can change the strategy in the subgame so that he will get higher reward against $\sigma_1^*$ which conflicts the definition of NE.

And we will further decompose $I_1^i \in S_{top}$ into two parts, $\{I_1^{i,-}\}$ and $\{I_1^{i,+}\}$. They have the property that $CBV^{\sigma_2^*}(I_1^{i,-}) - CBV^{\sigma_2^S}(I_1^{i,-}) \leq 0$ and $CBV^{\sigma_2^*}(I_1^{i,+}) - CBV^{\sigma_2^S}(I_1^{i,+}) > 0$. And since $\sum_i \tilde{p}(I_1^i)(CBV^{\sigma_2^*}(I_1^i) - CBV^{\sigma_2^S}(I_1^i)) \leq 0$ as discussed above, we have

$$\sum_{I_1^{i,-}} \tilde{p}(I_1^i)(CBV^{\sigma_2^*}(I_1^{i,-}) - CBV^{\sigma_2^S}(I_1^{i,-})) + \sum_{I_1^{i,+}} \tilde{p}(I_1^i)(CBV^{\sigma_2^*}(I_1^{i,+}) - CBV^{\sigma_2^S}(I_1^{i,+}))$$

$$= \sum_i \tilde{p}(I_1^i)(CBV^{\sigma_2^*}(I_1^i) - CBV^{\sigma_2^S}(I_1^i)) \tag{8}$$

$$\leq 0$$

which implies that

$$\sum_{I_1^{i,+}} \tilde{p}(I_1^i)(CBV^{\sigma_2^*}(I_1^{i,+}) - CBV^{\sigma_2^S}(I_1^{i,+})) \leq -\sum_{I_1^{i,-}} \tilde{p}(I_1^i)(CBV^{\sigma_2^*}(I_1^{i,-}) - CBV^{\sigma_2^S}(I_1^{i,-})) \tag{9}$$

Then we have

$$\sum_i (\hat{p}(I_1^i) - \tilde{p}(I_1^i))(CBV^{\sigma_2^*}(I_1^i) - CBV^{\sigma_2^S}(I_1^i))$$

$$= \sum_{I_1^{i,-}} (\hat{p}(I_1^{i,-}) - \tilde{p}(I_1^{i,-}))(CBV^{\sigma_2^*}(I_1^{i,-}) - CBV^{\sigma_2^S}(I_1^{i,-}))$$

$$+ \sum_{I_1^{i,+}} (\hat{p}(I_1^{i,+}) - \tilde{p}(I_1^{i,+}))(CBV^{\sigma_2^*}(I_1^{i,+}) - CBV^{\sigma_2^S}(I_1^{I_1^{i,+}}))$$

$$\leq \tau \Big( -\sum_{I_1^{i,-}} \tilde{p}(I_1^{i,-})(CBV^{\sigma_2^*}(I_1^{i,-}) - CBV^{\sigma_2^S}(I_1^{I_1^{i,-}})) \tag{10}$$

$$+ \sum_{I_1^{i,+}} \tilde{p}(I_1^{i,+})(CBV^{\sigma_2^*}(I_1^{i,+}) - CBV^{\sigma_2^S}(I_1^{I_1^{i,+}})) \Big)$$

$$\leq -2\tau \sum_{I_1^{i,-}} \tilde{p}(I_1^{i,-})(CBV^{\sigma_2^*}(I_1^{i,-}) - CBV^{\sigma_2^S}(I_1^{I_1^{i,-}}))$$

$$\leq -2\tau \min_{I_1^j} \Big( CBV^{\sigma_2^*}(I_1^j) - CBV^{\sigma_2^S}(I_1^j) \Big)$$

The last inequation holds since $\min_{I_1^j} \Big( CBV^{\sigma_2^*}(I_1^j) - CBV^{\sigma_2^S}(I_1^j) \Big) \leq 0$ since $\sigma_2^*$ is the strategy with lowest exploitability by only changing strategy of $\sigma_2$ in the subgames.

Back to Equation 7, we have

$$\sum_i \hat{p}(I_1^i)(v_1^\sigma(I_1^i) - CBV^{\sigma_2^S}(I_1^i)) \leq \Delta - 2\tau \min_{I_1^j} \Big( CBV^{\sigma_2^*}(I_1^j) - CBV^{\sigma_2^S}(I_1^j) \Big) \tag{11}$$

And substitute it into Equation 6,

$$\Delta + (1 - \alpha - 2\alpha\tau) \min_{I_1^j} \Big( CBV^{\sigma_2^*}(I_1^j) - CBV^{\sigma_2^S}(I_1^j) \Big)$$

$$\geq (1-\alpha) \min_{I_1^j} \Big( v_1^\sigma(I_1^j) - CBV^{\sigma_2^S}(I_1^j) \Big) + \alpha(\Delta - 2\tau) \min_{I_1^j} \Big( CBV^{\sigma_2^*}(I_1^j) - CBV^{\sigma_2^S}(I_1^j) \Big) \tag{12}$$

$$\geq -\Delta$$

so that

$$CBV^{\sigma_2^S}(I_1^j) \leq CBV^{\sigma_2^*}(I_1^j) + \frac{2}{1 - (2\tau + 1)\alpha}\Delta \tag{13}$$

for all $I_1^j$ in the subgame.

And for infoset $I$ out of the subgame with level 0, since the refined strategy $\sigma_2^S$ and blueprint strategy $\sigma_2$ are the same here, the $CBV$ value is exactly the same and the inequality holds.

INDUCTIVE STEP

The inductive step mostly follows that of Brown & Sandholm (2017).

Since $CBV^{\sigma_2^S}(I_1) \le CBV^{\sigma_2^*}(I_1) + \frac{2}{1-(2\tau+1)\alpha}\Delta$ holds for every subgame $S$, $\sigma_2'$ will also satisfy this inequation since $\Delta$ and $\tau$ are defined as maximum over all subgames.

Now, suppose $CBV^{\sigma_2'}(I_1) \le CBV^{\sigma_2^*}(I_1) + \frac{2}{1-(2\tau+1)\alpha}\Delta$ holds for any infoset with level lower or equal to $k$, we will prove that it also holds for infoset with level $k+1$.

By definition of $CBV(I_1)$,

$$
\begin{aligned}
CBV^{\sigma_2}(I_1,a) &= \Big( \sum_{h\in I_1} \pi_{-1}^{\sigma_2}(h) v^{\langle CBR(\sigma_2),\sigma_2\rangle}(h\cdot a)\Big)/\sum_{h\in I_1}\pi_{-1}^{\sigma_2}(h) \\
&= \Big( \sum_{h\in I_1}\pi_{-1}^{\sigma_2}(h) \sum_{h'\in succ(h,a)} \pi_{-1}^{\sigma_2}(h,h')v^{\langle CBR(\sigma_2),\sigma_2\rangle}(h')\Big)/\sum_{h\in I_1}\pi_{-1}^{\sigma_2}(h) \quad (14) \\
&= \Big( \sum_{h\in I_1}\sum_{h'\in succ(h,a)}\pi_{-1}^{\sigma_2}(h')v^{\langle CBR(\sigma_2),\sigma_2\rangle}(h')\Big)/\sum_{h\in I_1}\pi_{-1}^{\sigma_2}(h)
\end{aligned}
$$

We can swap the two summations above since the game is perfect recall, then

$$
CBV^{\sigma_2}(I_1,a) = \Big( \sum_{I_1'\in succ(I_1,a)}\sum_{h'\in I_1'}\pi_{-1}^{\sigma_2}(h')v^{\langle CBR(\sigma_2),\sigma_2\rangle}(h')\Big)/\sum_{h\in I_1}\pi_{-1}^{\sigma_2}(h) \quad (15)
$$

By substituting the definition of $CBV(I_1')$ into the equation above,

$$
CBV^{\sigma_2}(I_1,a) = \Big( \sum_{I_1'\in succ(I_1,a)}CBV^{\sigma_2}(I_1')\sum_{h'\in I_1'}\pi_{-1}^{\sigma_2}(h')\Big)/\sum_{h\in I_1}\pi_{-1}^{\sigma_2}(h) \quad (16)
$$

And by the induction hypothesis,

$$
CBV^{\sigma_2}(I_1,a) \le \Big( \sum_{I_1'\in succ(I_1,a)}(CBV^{\sigma_2^*}(I_1')+\frac{2-\alpha}{1-\alpha}\Delta)\sum_{h'\in I_1'}\pi_{-1}^{\sigma_2}(h')\Big)/\sum_{h\in I_1}\pi_{-1}^{\sigma_2}(h) \quad (17)
$$

Because $I_1$ is out of the subgame and $\sigma_2^*,\sigma_2$ is exactly the same outside the subgame, we will get

$$
\begin{aligned}
CBV^{\sigma_2}(I_1,a) &\le \Big( \sum_{I_1'\in succ(I_1,a)}(CBV^{\sigma_2^*}(I_1')+\frac{2-\alpha}{1-\alpha}\Delta)\sum_{h'\in I_1'}\pi_{-1}^{\sigma_2^*}(h')\Big)/\sum_{h\in I_1}\pi_{-1}^{\sigma_2^*}(h) \\
&= CBV^{\sigma_2^*}(I_1,a)+\frac{2-\alpha}{1-\alpha}\Delta\Big( \sum_{I_1'\in succ(I_1,a)}\sum_{h'\in I_1'}\pi_{-1}^{\sigma_2^*}(h')\Big)/\sum_{h\in I_1}\pi_{-1}^{\sigma_2^*}(h) \quad (18) \\
&= CBV^{\sigma_2^*}(I_1,a)+\frac{2-\alpha}{1-\alpha}\Delta
\end{aligned}
$$

Finally, by mathematical induction we get

$$
\exp(\sigma_2') \le \exp(\sigma_2^*)+\frac{2}{1-(2\tau+1)\alpha}\Delta \quad (19)
$$

## A.2 PROOF OF THEOREM 2

**Theorem 2.** (opponent exploitation) *Let* $\epsilon = \|\hat{p}-p\|_1$ *be the L1 distance of the distribution* $p(I_1^i)$ *and* $\hat{p}(I_1^i)$. *Let* $\eta = \min_{S\in\mathbb{S}}\max_{I_1^j\in S_{\text{top}}}\Big(CBV_1(I_1^j,\sigma_2^S)-CBV_1(I_1^j,\sigma_2^*)\Big) \ge 0$. *We use* $BR_p^{[\mathbb{S},\sigma_p]}(\sigma)$ *to denote the strategy for player* $p$ *which maximizes its utility in subgame* $S\in\mathbb{S}$ *against* $\sigma_{-p}$ *under the constraint that* $BR_p^{[\mathbb{S},\sigma_p]}(\sigma)$ *and* $\sigma_p$ *differs only inside* $\mathbb{S}$. *By maximizing objective 2, for all* $S\in\mathbb{S}$, *the refined strategy* $\sigma_2'$ *satisfies*

$$
u_2^{\left\langle BR_1^{[\mathbb{S},\sigma_1]}(\sigma_2'),\sigma_2'\right\rangle}(S) \ge u_2^{\left\langle BR_1^{[\mathbb{S},\sigma_1]}(\sigma_2^*),\sigma_2^*\right\rangle}(S)+\frac{1-\alpha}{\alpha}(\eta-2\Delta)-\epsilon\eta \quad (20)
$$

**Proof:** Still, we only consider a specific subgame $S$ first.

$\sigma_2^S$ is maximizing

$$(1 - \alpha) \underbrace{\min_{I_1^j} \left( v_1^\sigma(I_1^j) - CBV^{\sigma_2^S}(I_1^j) \right)}_{g(\sigma_2^S)} + \alpha \underbrace{\sum_i \hat{p}(I_1^i)(v_1^\sigma(I_1^i) - CBV^{\sigma_2^S}(I_1^i))}_{f(\sigma_2^S)} \tag{21}$$

So, we have

$$(1 - \alpha)g(\sigma_2^S) + \alpha f(\sigma_2^S) \geq (1 - \alpha)g(\sigma_2^*) + \alpha f(\sigma_2^*) \tag{22}$$

and

$$\begin{aligned}
&\max_{I_1^j} CBV(I_1^j, \sigma_2^S) - CBV(I_1^j, \sigma_2^*) = \eta \geq \eta \\
&\Leftrightarrow g(\sigma_2^S) - \Delta \leq -\eta \\
&\Leftrightarrow g(\sigma_2^S) - \Delta \leq \Delta + g(\sigma_2^*) - \eta \quad (g(\sigma_2^*) \geq -\Delta)
\end{aligned} \tag{23}$$

Therefore,

$$\alpha f(\sigma_2^S) \geq \alpha f(\sigma_2^*) + (1 - \alpha)(\eta - 2\Delta) \tag{24}$$

which means

$$\begin{aligned}
&\sum_i \hat{p}(I_1^i)(CBV^{\sigma_2^*}(I_1) - CBV^{\sigma_2^S}(I_1^i)) \\
&\geq \frac{1 - \alpha}{\alpha}(\eta - 2\Delta) + \sum_i \hat{p}(I_1^i)(CBV^{\sigma_2^*}(I_1) - CBV^{\sigma_2^*}(I_1^i)) \\
&\Leftrightarrow -\sum_i \hat{p}(I_1^i)CBV^{\sigma_2^S}(I_1^i) \geq \frac{1 - \alpha}{\alpha}(\eta - 2\Delta) - \sum_i \hat{p}(I_1^i)CBV^{\sigma_2^*}(I_1^i) \\
&\Leftrightarrow -\sum_i p(I_1^i)CBV^{\sigma_2^S}(I_1^i) \geq \frac{1 - \alpha}{\alpha}(\eta - 2\Delta) - \sum_i p(I_1^i)CBV^{\sigma_2^*}(I_1^i) \\
&\qquad\qquad\qquad - \sum_i (p(I_1^i) - \hat{p}(I_1^i))(CBV^{\sigma_2^S}(I_1^i) - CBV^{\sigma_2^*}(I_1^i)) \\
&\Rightarrow -\sum_i p(I_1^i)CBV^{\sigma_2^S}(I_1^i) \geq \frac{1 - \alpha}{\alpha}(\eta - 2\Delta) - \sum_i p(I_1^i)CBV^{\sigma_2^*}(I_1^i) - \epsilon\eta \\
&\Leftrightarrow \sum_i p(I_1^i)V_2(I_1^i, BR(\sigma_2^S), \sigma_2^S) \geq \frac{1 - \alpha}{\alpha}(\eta - 2\Delta) - \epsilon\eta + \sum_i p(I_1^i)V_2(I_1^i, BR(\sigma_2^*), \sigma_2^*) \\
&\Leftrightarrow u_2^{\langle BR_1^{[S,\sigma_1]}(\sigma_2^{[S \leftarrow \sigma_2^S]}), \sigma_2^{[S \leftarrow \sigma_2^S]} \rangle}(S) \geq u_2^{\langle BR_1^{[S,\sigma_1]}(\sigma_2^*), \sigma_2^* \rangle}(S) + \frac{1 - \alpha}{\alpha}(\eta - 2\Delta) - \epsilon\eta
\end{aligned} \tag{25}$$

Since $\eta$ is defined as minimum over all subgames, we have

$$u_2^{\left\langle BR_1^{[S,\sigma_1]}(\sigma_2'), \sigma_2' \right\rangle}(S) \geq u_2^{\left\langle BR_1^{[S,\sigma_1]}(\sigma_2^*), \sigma_2^* \right\rangle}(S) + \frac{1 - \alpha}{\alpha}(\eta - 2\Delta) - \epsilon\eta \tag{26}$$

## B   POKER RULES

### RULES OF LEDUC POKER

Leduc Poker is a two players game. In Leduc Poker, there are 6 cards in total, three ranks($\{J, Q, K\}$) with two suits($\{a, b\}$) each. And at the beginning, every player should put 1 chip into the pot and then will be dealt with one private card. Then, two players alternatively bet. They can call, raise and fold. If any of them fold, the game ends and all chips in the pot belongs to the other player. And

when a player call, he has to put chips in the pot to ensure that he contributes equal chips as the other player in the pot. When a player raise, he has to ensure that he contributes more chips than the other player in the pot. A betting round ends when a player calls.

Leduc Poker is divided into two betting rounds. In the first round, a private card is dealt to each player and then two player start to bet. After the first betting round ends and nobody folds, a public card is dealt on board and the second betting round starts. When the second round ends, both of the player show their private hands and the stronger hands win. If a player's private card has the same rank as the public card, then he wins. Otherwise, we have $J < Q < K$ and the higher one wins. And in each betting round, there will be at most two raises in our experiment and each raise should contribute 1 more chip in the first round and 2 more chips in the second round.

### RULES OF FLOP HOLD'EM POKER

The rules of Flop Hold'em Poker is similar to that of Leduc Poker. In FHP, we use the standard 52-card deck. At the beginning, the first player will contribute 1 chip to the pot and the second player will contribute 2 chips. And then they will be dealt with 2 private cards each and the first player start to bet. There are still two betting rounds and the raise sizes are both 2 chips. At the end of the first betting round, there will be 3 public cards dealt on board. And the players will show their private card at the end of the second betting round and the larger one wins the game. In FHP, we have the same rule of card order as a standard Texas hold 'em .

## C   IMPLEMENTATION DETAILS

**Leduc Poker.** In Leduc Poker, we solve for a blueprint strategy using a variant of CFR algorithm (Lanctot et al., 2009; Tammelin et al., 2015) with 1M iterations in the full game. Then we apply search in subgames when the board card is dealt.

**Flop Hold'em Poker (FHP).** As for FHP, there are 1,286,792 different infosets for each betting sequence. We cluster them into 200 infosets by an abstraction algorithm (Johanson et al., 2013) in order to make equilibrium finding feasible. Then, we compute a blueprint strategy in this abstraction with 10,000,000 iterations. We apply search immediately once the flop cards are dealt.

