# OpenReview forum: "Safe Opponent-Exploitation Subgame Refinement"
_ICLR.cc/2022/Conference — ICLR 2022 Submitted_

### Official Review · Reviewer_f5FT · 2021-10-25

**Correctness:** 4
**Technical Novelty And Significance:** 2
**Empirical Novelty And Significance:** 2
**Recommendation:** 3
**Confidence:** 4

**Main Review:**

Strengths:

1. The paper is clean and easy to read.

2. Both the theoretical results and the experiments clearly support the main claims of the paper.

Weaknesses:

1. The main contribution of the paper is only a minimal step on top of known ideas (a convex combination of unsafe and maxmargin subgame solving). The theoretical and experimental results are not at all surprising either.

2. Related to above: I'm not completely satisfied with the comparison to the past safe exploitation works [Johanson et al '07, Ganzfried & Sandholm '15]. As far as I can tell, the present paper simply uses a different parameterization of the Pareto frontier of exploitation vs exploitability--it uses the mixing amount $\alpha$; the others use the exploitability, formulating the objective as "maximize exploitation such that exploitability is at most $\varepsilon$". These are not very different. Further, LP solving in games is not that slow, especially with recent advances [Zhang & Sandholm '20]; in particular, the subgames in experiments seem tractable with LP.

[Zhang & Sandholm '20] Sparsified Linear Programming for Zero-Sum Equilibrium Finding

**Summary Of The Paper:**

The paper introduces Safe Exploitation Search (SES), which achieves a balance between maxmargin solving (guaranteed to be safe, but non-exploitative) and unsafe subgame solving (unsafe, as the name suggests, but exploitative of the opponent's initial distribution at S_top) by mixing their objective functions. The paper provides both theoretical and experimental evidence that the method does a good job of balancing the two objectives.

**Summary Of The Review:**

Well-written, but in my view not very novel/just incremental work over past work on subgame solving.

---

> ### Author Response · Authors · 2021-11-22
> **Response to Reviewer f5FT (1/2)**
>
> We appreciate the reviewer for the insightful comments.
>
> **Q1:** "I'm not completely satisfied with the comparison to the past safe exploitation works [Johanson et al '07, Ganzfried & Sandholm '15]. As far as I can tell, the present paper simply uses a different parameterization of the Pareto frontier of exploitation vs exploitability -- it uses the mixing amount $\alpha$; the others use the exploitability, formulating the objective as 'maximize exploitation such that exploitability is at most $\varepsilon$'. These are not very different."
>
> **A1:** We thank the reviewer to point out the comparions to [Johanson et al '07] and [Ganzfried & Sandholm '15]. **We have added detailed comparisons with previous safe exploitation algorithms in Section 2, and empirical results in Section 5.3 in the revision.**
>
> We agree with the reviewer that our optimizing objective is not very different from "maximize exploitation such that exploitability is at most $\varepsilon$". However, we respectfully insist that the real-time search framework of SES plays a vital role. In added experiments, SES is much more efficient than p-RNR [Johanson 2007]. The reason is that p-RNR is a ***"global"*** algorithm, which means that whenever the opponent model or the exploitation level ($p$ in p-RNR) changes, it has to recompute the strategy for the whole game. In contrast, SES only efficiently computes safe exploitation strategies in subgames encountered, which are much smaller than the whole game. To our best knowledge, SES is the first algorithm that implements both "$\varepsilon$-safe exploitation" and "online adaptation" to a possibly changing opponent model.
>
> The settings of [Ganzfried & Sandholm '15] and our paper are quite different. [Ganzfried & Sandholm '15] study strategies in ***"repeated games"***, and provide a full characterization of safe deviation from NE. In contrast, our work focuses on the safety of ***"stage game"*** strategies. The two definitions of "safety" are different.
>
> Furthermore, we find that our algorithm is complementary to RWYWE proposed in [Ganzfried & Sandholm '15]. RWYWE generally proceeds as follows:
>
> 1. For each timestep $t$ in a repeated game, plays a $k^t$-safe best response $\pi_t^t$ with respect to the estimated opponent's strategy.
> 2. $k^{t+1} = k^t +$ expected utility of $\pi_i^t$ against the observed opponent's action $-$ NE value $v^*$
>
> RWYWE achieves safety because it uses $k$ to track how much it has earned over NE in previous rounds, and then risk at most $k$ in exploitability. As stated in their paper, computing an $\varepsilon$-safe best response at every round is non-trivial. It is one of the main restrictions which limits RWYWE's scalability to even medium-sized games. If viewing SES as an efficient algorithm that approximates such $\varepsilon$-safe exploitation strategy, it is possible to integrate  SES into RWYWE, and provide more scalable algorithms for safe exploitation in repeated games as well.
>
> **Q2:** "Further, LP solving in games is not that slow, especially with recent advances [Zhang & Sandholm '20]; in particular, the subgames in experiments seem tractable with LP."
>
> **A2:** We admit that using LP to solve games in our experiments is feasible. However, as stated in A1,  opponent exploitation is an "online process" where an algorithm should adaptively respond at each round. Previous works [Johanson et al '07, Ganzfried & Sandholm '15] need to recompute a new $\varepsilon$-safe best response for the **whole** game at every round. It is still computationally hard for LP and CFR without real-time search. The added experiment in revision supports our argument that [Johanson et al '07] is much less efficient than our algorithm. From the other perspective, it is possible to use LP in the computation of SES, by using LP instead of CFR to solve the gadget game constructed by SES. Real-time search algorithms usually do not restrict which solver to use, and this is also true for SES.
>
> What's more, in practice (especially in the real-time search domain), we do not require an extremely accurate Nash equilibrium. Usually, $10mbb/h$($0.01bb/h$) is a very accurate Nash equilibrium in practice. And [Zhang & Sandholm '20] also illustrate that Poker Specific DCFR is much faster than the best LP solver when we only require a strategy with exploitability of no more than $0.01$. DCFR only takes no more than 10 CPU seconds in River Endgame Solving while LP solver takes more than 100 CPU seconds.

---

> > ### Author Response · Authors · 2021-11-22
> > **Response to Reviewer f5FT (2/2)**
> >
> > **Q3:** "The main contribution of the paper is only a minimal step on top of known ideas (a convex combination of unsafe and maxmargin subgame solving)." "Not very novel/just incremental work over past work on subgame solving."
> >
> > **A3**: Summing up A1 and A2, in spite of its simplicity in form, to our best knowledge, SES is the first algorithm that can efficiently perform safe exploitation in an "online" setting, where the opponent model is updated with streamed data. It relieves a serious restriction on practical use cases of previous algorithms. We also prove explicit bounds for both safety and exploitation, taking into account various practical factors such as opponent modeling error and blueprint strength. Our added experiments show that, with the aid of real-time search, SES is significantly more efficient than previous algorithm [Johanson et al '07] without search. Therefore, we respectfully insist that the advantages of our algorithm over previous works and its significance should not be underestimated.

---

> > ### Comment · Reviewer_f5FT · 2021-11-25
> > **Response to response**
> >
> > I feel that the response has only caused me to have more questions rather than fewer.
> >
> > 1. You claim that RNR is orders of magnitude less efficient computationally than SES. This is only possible if SES is used only in subgames "late" in the tree. I am assuming that this is what you are doing--in any case, you should specify explicitly in the paper how the subgames are defined in the experiments. Therefore, in particular, the early-game strategy is not exploitative at all, but rather identical to the blueprint, correct? I was previously (incorrectly, apparently) thinking of SES as being applied at every decision point in a nested fashion (as is common in the subgame solving literature).
> >
> > 1. Speaking of comparison to RNR, why is the opponent model used only for root probabilities, and not for modeling strategies during the subgame itself? Indeed, this seems like the only difference between SES and "RNR applied to the maxmargin gadget subgame". The latter would be an alternative to SES that carries over the theoretical guarantees of RNR in a more direct way. It seems like a strong baseline that should be compared against, if nothing else.
> >
> > 1. Building off the discussion with SWZo: My understanding of the revised Thm 1 is as follows. To give a nontrivial exploitability guarantee, it must be the case that $\hat p$ applies zero probability wherever $\tilde p$ applies zero probability, correct? (Otherwise, $\tau$ is infinite) (You should probably specify explicitly that the max in the definition of $\tau$ is only taken over $I^i_1$ such that $\tilde p(I^i_1) > 0$, which I'm assuming is actually the case; if this is not the case then we have bigger problems.) This seems like a very strong condition, especially considering that, in some sense intuitively, the mistakes generated by playing actions that are zero probability in equilibrium are exactly the "worst" mistakes, that should be exploited the most. In light of (2) above, this means in my mind that the safety guarantee of SES will be often simply trivial.
> >
> > 4. Why does RNR have worse performance than SES on all metrics? In particular, RNR(p=1) should be just computing a best response, and RNR(p=0) should be computing an NE, so they should be at least as good as SES(α=1) and SES(α=0) on the evaluation and exploitability metrics respectively. I assume this is only an artifact of RNR (even RNR(big)) not being given enough time to converge. I feel that the authors should make this more clear, as in that case the right takeaway is "SES is more computationally efficient than RNR" instead of "SES converges to better exploitation/exploitability than RNR". Indeed, the latter would be very surprising in my opinion if it were true.

---

### Official Review · Reviewer_SWZo · 2021-10-28

**Correctness:** 3
**Technical Novelty And Significance:** 3
**Empirical Novelty And Significance:** 4
**Recommendation:** 8
**Confidence:** 4

**Main Review:**

The main idea of the submission is somewhat obvious. However, I think it nevertheless offers value to the community, as opponent exploration is both an important and an under-explored research area (at least in my opinion). There are some small problems with the submission (see below) that I would like to see addressed. More generally, I like to see the authors put into the quality of the writing. As it currently stands, much of it is sloppy as does not read like a publication-ready conference paper.

### Exploitability Theory
I think the exploitability results are a bit confusing as currently written. The term "exploitability" is not generally used in a way that depends on the opponent's "true distribution". Exploitability defines a worst case loss. Unless I am misunderstanding something, the submission is talking about a worst case loss, subject to a constraint on the private information. I am not sure it is appropriate to refer to this quantity as exploitability --- SES is not, in fact, safe against arbitrary opponents.

The submission provides a nice example illustrating why its approach is superior to simply mixing between a safe strategy and an exploitation strategy. That being said, it seems to me to be perfectly possible to provide theorems analogous to Theorem 1 and Theorem 2 for the "mixing between a safe strategy and an exploitation strategy" approach. I think it would add value to the submission to work through these results, as it may be interesting to compare them to Theorem 1 and Theorem 2.


### Baselines
The submission dismisses Johanson 2007 and Ganzfried 2015 as baselines. However, I am not sure that this dismissal is well justified. The method in the submission already uses an estimate of the opponent strategy. It would be relatively to substitute this estimate for the "known" component of the strategy required by restricted Nash response (RNR) and test the exploitability. It may be that RNR performs perfectly well in this case. Also, while I do not dispute it is difficult to scale LP solvers to large games, the games considered in the submission are not that large. Therefore, it seems like both Johanson and Ganzfried offer reasonable baselines for the algorithm proposed in the submission.

### Small Problems

Paper cites benchmark Hanabi paper for superhuman Hanabi performance?

Paper cites Tian for superhuman bridge performance?

“Monte Carlo tree search (MCTS) can be viewed as a powerful policy improvement operator.”
This sentence appears to be (hopefully accidentally) plagiarized from the AlphaGo Zero paper: “MCTS may therefore be viewed as a powerful policy improvement operator”.

Poorly constructed “They both have sounded monotonic improvement guarantees in theory, and show tremendous performance promotion empirically.”

“In fully-cooperative imperfect information games, search algorithms proposed in Tian et al. (2020b) and Lerer et al. (2020) are also proved to never be detrimental to the current policy”
This is only half true. Lerer et al.’s “never detrimental” is only asymptotic. After any finite amount of time (ie in all practical cases) there is no such guarantee.

“However, it may be overly conservative confronted with opponents with limited rationality and fail to take advantages of their weaknesses to obtain higher rewards.”
It has nothing to reference (should be they)
Advantages is wrong grammatical number

“We construct a new gadget game to optimize this objective, which enable”

“Search techniques are proven to be essential in developing strong AI strategies”
This claim appears overly strong.

“Libratus (Brown et al., 2018) is the first superhuman poker AI which is considered the milestone in the research of imperfect information games”
This sentence is both grammatically incorrect and unnecessarily contentious. Does it add value to the paper to make a claim about which poker AI algorithm (between Libratus and DeepStack) was the first and which is considered the “milestone”? Also, is it appropriate to not even cite DeepStack here?

“search algorithms for subgame refinement have also shown promise in improving joint strategies in cooperative imperfect information games such as Hanabi (Bard et al., 2020)”
The paper being cited here does not involve sublime refinement.

“In that paper, Johanson et al. (2007) assumes that the”
assume

“But Ganzfried & Sandholm (2015a) relies on LP solver”
rely

“It is worthwhile mentioning that there also have been extensive research”
has

“However, this paper only only focuses”
fix

Notation Section:
It’s bad form to start sentences with symbols.

It’s pretty unusual for EFGs to use \sqsubseteq the way the submission defines it.
Having read further it appears that the submission uses \sqsubseteq the way it is normally used for EFGs. So the definition is incorrect.

In the matrix game shown in Table 1, there are two NEs.
Any interpolation of the two NE the submission gives is also a NE.

Throughout the paper Safe Subgame Exploitation is improperly capitalized.

**Summary Of The Paper:**

The submission proposes performing safe opponent exploitation via decision-time planning using a novel gadget game. At the top of the gadget game, there is a chance node that goes to the maxmargin gadget game with probability (1 - alpha) and goes to the estimate posterior over private information with probability alpha. The submission proves that, for small enough alpha and posterior error, this gadget game produces a safe (non-exploitable) strategy. The submission presents experiments on Leduc Hold'em and Flop Hold'em Poker.

**Summary Of The Review:**

I would like the authors to address the comments above. Should they do so, I am willing to reconsider my score.

---

After discussions with the authors, I raised my score from a 5 to an 8.

---

> ### Author Response · Authors · 2021-11-22
> **Response to Reviewer SWZo (1/2)**
>
> We appreciate the reviewer for the insightful comments. Especially, we are grateful to the reviewer for pointing out typos in writing and citations, and we have made corrections accordingly in the revision. We provide clarification to address other concerns of the reviewer in the following.
>
> **Q1**: "The term 'exploitability' is not generally used in a way that depends on the opponent's 'true distribution'." "SES is not, in fact, safe against arbitrary opponents."
>
> **A1**: We follow the common definition of exploitability in this paper. We find that the original statement of Theorem 1 may cause the reviewer's confusion. We have refined this statement. In Theorem 1, the exploitability does not depend on the distribution of the best response. $\tau$ indicates how much $\hat p$ deviates from the distribution given by a Nash equilibrium $\langle \sigma_1^*, \sigma_2^* \rangle $ where $\sigma_2^*$ is constrained to be the same with $\sigma_2$ outside the subgame. Therefore, $\tau$ is a fixed value no matter what the best response is.
>
> **Q2**: "That being said, it seems to me to be perfectly possible to provide theorems analogous to Theorem 1 and Theorem 2 for the 'mixing between a safe strategy and an exploitation strategy' approach."
>
> **A2**: We provide both exploitability and expected utility results for mixing strategy. Let ${\rm mix}_p(\sigma_1, \sigma_1')$ be the mixing strategy which plays $\sigma_1$ with probability $p$, and $\sigma_1'$ with $1-p$. [Johanson 2007] has already provided a full characterization of its expected utility: $u({\rm mix}_p(\sigma_1, \sigma_1'), \sigma_2)=pu(\sigma_1, \sigma_2) + (1-p)u(\sigma_1', \sigma_2)$. As for exploitability, it can be shown that $\exp({\rm mix}_p(\sigma_1, \sigma_1'))\leq p\exp(\sigma_1)+(1-p)\exp(\sigma_1')$. The example in Figure 2 shows that the bound is tight.
>
> In some circumstances, $\exp(\sigma_{unsafe})$ can be quite large. Therefore, the exploitability of the mixing strategy can also be large even for a rather small $p$. In contrast, as shown in Theorem 1, SES's exploitability does not rely on $\exp(\sigma_{unsafe})$, which makes our bound for SES tighter than that of mixed strategy in most cases.
>
> **Q3**: "The submission dismisses Johanson 2007 and Ganzfried 2015 as baselines. However, I am not sure that this dismissal is well justified."
>
> **A3**: As suggested by the reviewer, **we have added detailed comparisons with previous safe exploitation algorithms in Section 2, and empirical results in Section 5.3 in the revision.**
>
> Experiments show that our algorithm is much more efficient than p-RNR proposed in [Johanson 2007]. p-RNR is a ***"global"*** algorithm, which means that whenever the opponent model or the exploitation level ($p$ in p-RNR) changes, it has to recompute the strategy for the whole game. In contrast, SES only needs to compute the whole strategy once, which is the blueprint. Thereafter, SES computes safe exploitation strategies in the encountered subgames, which are much smaller than the whole game. Consider a practical setting where opponent models are updated with new interaction data after each round. The efficiency of making online adaptations becomes important. Furthermore, in theory, [Johanson 2007] does not provide an explicit bound on exploitability, while Theorem 1 in our paper quantitatively takes into account some vital factors.
>
> We have to clarify that there are major differences in the settings of [Ganzfried & Sandholm '15] and our paper. [Ganzfried & Sandholm '15] study strategies in ***"repeated games"***. In contrast, our work is aimed at finding approximately safe ***"stage game"*** exploitation strategies. The two definitions of "safety" are different.
>
> Moreover, we find that our algorithm is complementary to RWYWE proposed in [Ganzfried & Sandholm '15]. RWYWE achieves safety by tracking how much it has earned over NE in previous rounds and risking at most this amount in exploitability. As stated in their paper, computing an $\varepsilon$-safe best response at every round is non-trivial. It is one of the main restrictions which limits RWYWE's scalability to even medium-sized games. If viewing SES as an approximate algorithm for calculating such $\varepsilon$-safe exploitation strategy, it is possible to integrate SES into [Ganzfried & Sandholm '15] and to provide more scalable algorithms for repeated games as well.

---

> > ### Author Response · Authors · 2021-11-22
> > **Response to Reviewer SWZo (2/2)**
> >
> > **Q4**: Also, while I do not dispute it is difficult to scale LP solvers to large games, the games considered in the submission are not that large.
> >
> > **A4**: We admit that using LP to solve games in our experiments is feasible. However, the opponent exploitation setting is more than calculating a strategy once. Instead, it requires calculating a responding strategy with respect to a possibly changing opponent model at each round. [Ganzfried & Sandholm '15] uses LP to find $k^t$-safe best response at each round. Solving the whole game over and over again is still computationally hard. As stated in A3, [Johanson 2007] is also a global algorithm that is much less efficient than SES. In contrast, SES does not need to recompute a whole game strategy. Our added experiments in section 5.3 support this argument. Therefore, SES suffices the need for online playing against opponents. From the other perspective, it is possible to use LP in the computation of SES, by using LP instead of CFR to solve the gadget game constructed by SES. Real-time search algorithms usually do not restrict which solver to use, and this is also true for SES.

---

> > > ### Comment · Reviewer_SWZo · 2021-11-24
> > > **Follow Up**
> > >
> > > > We follow the common definition of exploitability in this paper.
> > >
> > > I agree that the revised version uses the common definition of exploitability. I also agree that the statement of Theorem 1 was the source of my confusion. That being said, unless I am mistaken, this appears to have been a mistake on the part of the writing, rather than on the part of my interpretation. I have listed the point of confusion below for convenience. Please let me know if you feel that my interpretation was unfair.
> > >
> > > The original submission uses the definitions:
> > >
> > >  - Let $\tilde{p}(I_1^i)$ be the real reach probability,  and $\hat{p}(I_1^i)$ be its estimation.
> > >
> > > The revised submission uses the definitions:
> > >
> > >  - Let $\tilde{p}(I^i_1)$ be the reach probability given by $σ_1^*$. Let $\hat{p}(I_1^i)$ be the estimation of reach probability $p(I^i_1)$ given by the real opponent strategy.
> > >
> > > While the statement has been clarified in the revision, that one of the two main theorems was misstated signals some amount of sloppiness, which makes me less inclined to trust the submission's proofs, which I have not checked in full detail.
> > >
> > > > We provide both exploitability and expected utility results for mixing strategy.
> > >
> > > Great, it may be nice to emphasize this in the submission.
> > >
> > > > As suggested by the reviewer, we have added detailed comparisons with previous safe exploitation algorithms in Section 2, and empirical results in Section 5.3 in the revision.
> > >
> > > Great, thanks for doing this.
> > >
> > > Looking at these experiments, it seems like there are some questions regarding how to fairly compare an exploitation algorithm that uses subgame resolving to an exploitation algorithm that does not. This leads me to suggest two additional baselines. 1) Run RNR on top of the maxmargin gadget games; 2) Run the existing SES gadget game, but with the modification that $p_1$ is forced to play their blueprint strategy in $S_2$ (as notated in Figure 1. I am aware there is not much additional time to perform these experiments, but hope that these would only require small changes to the author's existing codebase to implement.

---

> ### Author Response · Authors · 2021-11-29
> **Thanks for raising the score to 8!**
>
> We would like to thank the reviewer for raising the score! We really appreciate the valuable comments and suggestions from the reviewer, which greatly help improve our work.

---

### Official Review · Reviewer_hmTT · 2021-11-02

**Correctness:** 4
**Technical Novelty And Significance:** 2
**Empirical Novelty And Significance:** 2
**Recommendation:** 5
**Confidence:** 4

**Main Review:**

To my understanding, the technique presented by the authors to interpolate between a Nash equilibrium strategy and a best response (say, for Player 1) simply boils down to the following:
- introduce a parameter $\alpha$ and optimize the objective $f(x) = (1-\alpha) \cdot \min_{y\in Y} \\{x^\top U y\\} + \alpha\cdot x^\top U \bar{y}$, where $x$ is a strategy of Player 1, $Y$ is the set of strategies of Player 2, and $\bar{y}$ is a "predicted" strategy for Player 2;
- solve the above optimization point (that is, solve $\max_{x \in X} f(x)$ where $X$ is the set of strategies of Player 1.
The presence of search seems just an implementation detail that obscures the above, straightforward idea.

If my understanding were to be confirmed correct, I find the paper very light theory-wise, and I definitely think that the idea should be stated for what it is instead of interspersing it with the search formalism, which ends up obscuring the idea and making it looks significantly more complex than it is.

Again assuming I didn't miss anything about the fundamental underlying idea of the paper, the technical results (Theorems 1 & 2) are not surprising at all, and just amount to bounding the two terms in the definition of $f(x)$ independently.

Given the limited technical contribution I would be significantly more satisfied with the paper if it had shown more compelling/insightful experimental evidence of the benefit of the approach. However, the paper only experiments on one domain (poker), and the experiments do not seem to show any particular interesting insight. For example, the experiments find that if the prediction $\bar{y}$ is accurate, then for large $\alpha$ the maximum of $f$ will be a best response to the strategy of the opponent, and if the prediction is not accurate, then one would be better off by picking a lower $\alpha$. I find that completely expected, but perhaps I missed some more subtle point?

**Summary Of The Paper:**

The paper proposes a technique to smoothly interpolate between computing a Nash equilibrium strategy and computing a best response. The technique is presented in the form of search for an extensive-form imperfect information game. Experiments on two poker variants are presented showing that the technique can indeed interpolate between the Nash equilibrium strategy ("safety") and the best response ("exploitation").

**Summary Of The Review:**

My biggest concern with the paper is about the significance. I believe that the idea behind the paper is simply optimizing a convex combination of functions, and that the search formalism that was imposed on top is mostly an implementation detail from a theory point of view. I also find the experiments unsurprising and not particularly insightful. Other than that, I do not believe the paper to be technically flawed or problematic in its claims. However, I find the significance concern is serious enough to prevent me from recommending acceptance. I look forward to a robust discussion with the authors on that point.

---

> ### Author Response · Authors · 2021-11-22
> **Response to Reviewer hmTT (1/2)**
>
> We appreciate the reviewer for the insightful comments.
>
> **Q1**: "The technique presented by the authors to interpolate between a Nash equilibrium strategy and a best response (say, for Player 1) simply boils down to the following: introduce a parameter $\alpha$ and optimize the objective $f(x)=(1-\alpha)\cdot \min_{y\in Y}x^TUy + \alpha x^TU\bar{y}$." "The presence of search seems just an implementation detail that obscures the above, straightforward idea."
>
> **A1**: Technically, because SES uses search in subgames, the exact formulation is more complex than $f(x)=(1-\alpha)\cdot \min_{y\in Y}x^TUy + \alpha x^TU\bar{y}$. The main reason is that a subgame is **not** an imperfect information game. An imperfect information game has a **unique** root. But in a subgame, there are multiple roots (with the same public information). We give the exact formulation in the following.
>
> $
> \max_{v,x}~(1-\alpha) m + \alpha \hat p^Tv~~~~~(1)
> $
>
> $
> s.t.~~~~~~v_I-m\geq CBV_1^{\sigma^S_2}(I)~~~~~(2)
> $
>
> $
> Ex=e~~~~~(3)
> $
>
> $
> F^T v-A_2^T x\leq 0~~~~~(4)
> $
>
> $
> x\geq 0~~~~~(5)
> $
>
> The notations are borrowed from [Moravcik et al '16]. In the equation above, $m$ denotes the margin $\min_{I_1^j\in S_{\rm top}} \Big(v_1^{\sigma}(I_1^j)-CBV_1^{\sigma_2^S}(I_1^j)\Big)$ and $v_j=v_1^{\sigma}(I_1^j)$. And in the constraint, $x$ denotes the probability of reaching each state in the subgame. And $Ex=e$ constraints that the probability of reaching a state is equal to the probability of going out of it. And (4) is the constraint to guarantee that $x$ is achieved when the opponent best responds to our strategy.
>
> We admit that our optimization objective for subgame search is a convex combination of exploitation and safety, but it is simple yet efficient and effective. This combination form is a commonly applied objective in safe opponent exploitation literature. As pointed out by another reviewer, this objective is closely related to "maximize exploitation such that exploitability is at most $\varepsilon$", which is used in [McCracken et al  04, Johanson et al '07].
>
> We respectfully disagree with the point that "search seems just an implementation detail that obscures the above, straightforward idea". **We have added detailed comparisons with previous safe exploitation algorithms in Section 2, and empirical results in Section 5.3 in the revision.** Our algorithm performs better and is much more efficient. Whenever the opponent model is updated with new interaction data, the previous algorithm [Johanson et al '07] has to recompute an $\varepsilon$-safe best response in the whole game, which is computationally inefficient. In contrast, SES only searches for strategies in subgames encountered. The real-time search framework enables SES to make efficient online adaptations with respect to a possibly changing opponent model, which is generally infeasible for previous works. As stated in [Ganzfried et al '15], online calculation of safe exploitation is one of the main limitations on the algorithm's scalability. We respectfully insist that the novelty and significance of our paper should not be underestimated.

---

> > ### Author Response · Authors · 2021-11-22
> > **Response to Reviewer hmTT (2/2)**
> >
> > **Q2**: "The technical results (Theorems 1 & 2) are not surprising at all, and just amount to bounding the two terms in the definition of $f(x)$ independently."
> >
> > **A2**: We respectfully disagree with this claim on the significance of the theoretical results in this work. To our best knowledge, SES is the first to provide explicit bounds on both safety and exploitation, taking into account various practical factors such as opponent modeling error and blueprint strength. Besides, we find that Theorem 1 can possibly answer some fundamental problems insufficiently addressed by previous works. [Brown et al '17] found that unsafe search usually counterintuitively achieves rather low exploitability but they did not explain the reason. We believe that our Theorem 1 partially explains why unsafe search is usually safe. When testing the unsafe search, [Brown et al '17] use their blueprint strategy to calculate $\hat p$, which is close to $\tilde p$ created by $\sigma^*$. Here $\sigma^*=\langle \sigma_1^*, \sigma_2^* \rangle$ is a Nash equilibrium when constrained $\sigma_2^*$ to have same strategy as $\sigma_2$ outside the subgame. As a result, $\tau$ is small and the exploitability is low.
> >
> > [McCracken et al '04] Peter McCracken and Michael Bowling. Safe strategies for agent modeling in games. Papers from the 2004 AAAI Fall Symposium.
> >
> > [Johanson et al '07] Michael Johanson, Michael Bowling, and Martin Zinkevich. Computing robust counter-strategies. 2007.
> >
> > [Ganzfried et al '15] Sam Ganzfried and Tuomas Sandholm. Safe opponent exploitation. *ACM Transactions on Economics and Computation (TEAC)*, 3(2):1–28, 2015a.
> >
> > [Moravcik et al '16] Moravcik M, Schmid M, Ha K, et al. Refining subgames in large imperfect information games. Proceedings of the AAAI Conference on Artificial Intelligence. 2016, 30.
> >
> > [Brown et al '17] Brown N, Sandholm T. Safe and nested endgame solving for imperfect-information games. Workshops at the thirty-first AAAI conference on artificial intelligence, 2017.

---

> ### Author Response · Authors · 2021-11-29
> **Thank you for your time and efforts in reviewing our work!**
>
> Dear Reviewer,
>
> Thank you for your time and efforts in reviewing our work. We have provided detailed clarification to address the issues raised in your comments. If our response has addressed your concerns, we would be grateful if you could re-evaluate our work.
>
> If you have any additional questions or comments, we would be happy to have further discussions.
>
> Thanks,
>
> The authors

---

> > ### Comment · Reviewer_hmTT · 2021-11-30
> > **Thank you!**
> >
> > Thank you for your responses on many of the points that my review raised. Your feedback has helped me clarify a few aspects. Though, since my understanding of the underlying convex combination structure was fundamentally correct, I still somewhat stand behind my comment that, given the exceptional simplicity of the idea I would have preferred if the authors had done more to convince that the approach is robust experimentally, and had presented a significantly more thorough experimental evaluation. However, your clarifications and to the revised version you have uploaded have improved my opinion of the paper, and we are now discussing internally.
> >
> > PS. As a minor comment that in no way affects the discussion about the paper, I have noticed that you are using the notation $\langle a, b \rangle$ to denote a pair $(a, b)$ in the revised version of the paper. As the $\langle\cdot,\cdot\rangle$ notation is (in my experience universally, as in, across disciplines) standard for the inner product of $a$ and $b$, I think you might want to pick a different symbol for tuples/pairs. As a final minor comment, I think that denoting the utility of a pair of strategies $(a,b)$ as $u^{\langle a,b\rangle}$ is very uncommon and serves as an unnecessary barrier of entry for your paper.

---

> > > ### Author Response · Authors · 2021-11-30
> > > **Response to Reviewer hmTT**
> > >
> > >
> > >
> > > Thank you very much for your feedback! We are pleased that our response has clarified your concerns. During the discussions with other reviewers, we have provided additional experimental results to more thoroughly evaluate our method.
> > >
> > > - Reviewer SWZo and f5FT proposed a new method EXP-STRATEGY. EXP-STRATEGY builds on the same gadget game of SES (Figure 1 in this paper). It is different from SES in that, it keeps opponent strategy fixed to the estimation in the right part of the gadget game while SES does not. SES exploits an estimated opponent infoset distribution, while EXP-STRATEGY exploits a full estimated opponent strategy.
> > > - As required by reviewer SWZo, we have done some comparisons between SES and EXP-STRATEGY in Leduc and FHP, confronted with opponents with different strengths ($\Pr_{shuffle}$), and we have committed to do more (our commitments are in a direct comment to this paper). In these experiments, we find that in some cases, SES can achieve lower exploitability than EXP-STRATEGY under the same evaluation results (experiment on FHP with opponent  $Pr_{shuffle}=0.3$). And EXP-STRATEGY usually has a wider range of evaluation performances than SES. Please refer to our 2nd&3rd responses to reviewer SWZo for details.
> > > - As SWZo has pointed out, since EXP-STRATEGY is also a new algorithm, which can be regarded as a variant of SES, so the intention of this paper is not to "argue that SES is strictly superior". We believe the main contribution of this paper is that it is the first safe exploitation algorithm that builds on real-time search, which enables computationally efficient online adaptations
> > >
> > >
> > > What's more, we are grateful for your suggestion on notations. We will refine them in the final revision.
> > >
> > > Thank you again for your time and efforts in reviewing our work!
> > >
> > > Best Regards,
> > >
> > > The authors

---

### Official Review · Reviewer_Wp7Z · 2021-11-03

**Correctness:** 3
**Technical Novelty And Significance:** 3
**Empirical Novelty And Significance:** 2
**Recommendation:** 6
**Confidence:** 3

**Main Review:**

This paper’s motivation is easy to follow and does a good job of introducing subgame search and opponent exploitation. I have not found any glaring errors in derivations or proofs. In terms of significance, this paper answers the natural question of how opponent weaknesses can be exploited without overly sacrificing safety. Experiments are sufficient.

One area this paper could do a lot better in terms of clarity. For example, the definition of CBR at the end of section 3.1 could be much more precise---the argument of CBR need only depend on player -p, and \sigma_p in the definition should really be CBR_p(\sigma_{-p}). In other areas, the authors could be more precise, e.g., in section 4.1, “under the assumption that the reach probabilities *remain the same*”, could be changed to “is \hat{p}(...)”. This is to disambiguate between \tilde{p} and \hat{p}. The gadget game presented seems correct to me, and Figure 1 was very useful in illustrating it. However, for readers unfamiliar with gadgets, the textual representation could be insufficient. For example, P2’s infosets should stretch over both the left and the right branches (which is briefly alluded to in point 1).

Theorem 1 provides a natural extension of the bounds of Moravcik et. al. One concern I have is how useful these bounds are in practice, in particular for selecting an appropriate value of alpha. In larger games (e.g., FHP), it is very likely that some information sets are never reached in practice even over many rounds of play. Hence, tau is not known, or may not even be estimated using data (this admittedly is a problem for any type of opponent modeling). The authors’ claim at the end of section 5: “In case of a bad estimation, we can always choose smaller \alpha to ensure safety”. In practice, how do we know we have a bad estimation if there are no samples to compare with?

The authors present experimental results with varying epsilon (estimation error of opponent strategy). What was the procedure in which this error was introduced?



**Summary Of The Paper:**

This paper proposes a method of subgame resolving in zero-sum extensive form games, blending in elements of theoretical safety, as well as more classical approaches based on opponent modeling. The method is fairly straightforward, utilizing a modified objective balancing between the safety objective (used in Moravcik et al) and a best-response like objective use in opponent modeling. To facilitate compatibility with modern solvers such as CFR, the authors propose gadgets transforming the refinement step into another zero-sum extensive form game.

**Summary Of The Review:**

I vote for this paper to be accepted. Apart from a lack of polish, there are no glaring errors in the paper, and the key ideas in the paper are easy to follow and appreciate.

---

> ### Author Response · Authors · 2021-11-22
> **Response to Reviewer Wp7Z**
>
> We appreciate the reviewer for the insightful comments. We are also grateful to the reviewer for pointing out clarity in some definitions and descriptions, and we have addressed these issues in the revision. We also provide clarification to the reviewer's main concerns in the following.
>
> **Q1**: "How useful these bounds introduced in the theorems are in practice, in particular for selecting an appropriate value of alpha?" "In practice, how do we know we have a bad estimation if there are no samples to compare with?"
>
> **A1**: We agree with the reviewer that accurately estimating the opponent modeling error ($\epsilon$ in our paper) sometimes can be challenging for any algorithms which rely on opponent modeling. Instead of calculating $\epsilon$, we can use online learning algorithms (e.g., UCB1, Exp3) to choose among a fixed set of alphas based on online evaluation performances. This technique is also introduced in previous exploitation papers [Johanson et al' 07, Ganzfried et al' 15].
>
> **Q2**: "How did the author produce opponents with different strengths in the experiment?"
>
> **A2**: Following [Brown et al '18], we enumerate every infoset in the blueprint strategy and shift the action distribution randomly with probability $Pr_{shuffle}$ = 0.3 or 0.7. In shifting, we multiply the probability of each action by a random value sampled from $Uniform(0, 1)$, and then re-normalize the probability distribution.
>
>
>
> [Johanson et al '07] Michael Johanson, Michael Bowling, and Martin Zinkevich. Computing robust counter-strategies. 2007.
>
> [Ganzfried et al '15] Sam Ganzfried and Tuomas Sandholm. Safe opponent exploitation. *ACM Transactions on Economics and Computation (TEAC)*, 3(2):1–28, 2015a.
>
> [Brown et al '18] Brown N, Sandholm T, Amos B. Depth-limited solving for imperfect-information games[J]. arXiv preprint arXiv:1805.08195

---

> > ### Comment · Reviewer_Wp7Z · 2021-11-27
> > **Follow-up**
> >
> > Thank you for the response.
> >
> > Regarding Q2:
> > I was not referring to generating opponents with different strengths, but rather the error in estimated opponent strategy (whose magnitude is governed by $\epsilon$, and each curve in a single plot in Fig 3). It probably isn't a huge deal, but should be included for completeness.
> >
> > Regarding connections to [Ganzfried & Sandholm '15] and [Johanson et al '07] raised by other reviewers:
> > I think an unnecessary amount of confusion was caused (at least to me) by referencing over-loaded terms with different meanings from these 2 papers. For example, the term "Gift" was used in page 7, when more easily understood terms like exploitable or suboptimal could be used. Given that the notion of safety in SES is is quite different from [Ganzfried & Sandholm '15], I am not sure if these references help with clarity.
> >
> > Regarding [Johanson et al '07]. One of the advantages SES holds is its requirement that only subgames are solved, as opposed to the full game. How does the objective between the 2 differ if say, there is only one subgame (i.e., the full game)?
> >
> > Having re-read the other comments and the paper, I felt that one of the advantages touted by SES ---  being able to adapt to a possibly changing opponent model, was rather poorly elaborated on. In its current state, it appears to be a justification to dismiss a comparison with RNR. While I agree with sentiment in theory, I cannot think of a reasonable scenario where this advantage is significant.

---

### Author Response · Authors · 2021-11-29
**Commitments**



We are grateful to all reviewers for their beneficial comments and advice on our paper. Especially, reviewer SWZo and f5FT propose an interesting method that is closely related to our algorithm SES. The method, which we call EXP-STRATEGY, builds on the same gadget game of SES (Figure 1 in this paper). It is different from SES in that, it keeps opponent strategy fixed to the estimation in the right part of the gadget game while SES does not. SES exploits an estimated opponent infoset distribution, while EXP-STRATEGY exploits a full estimated opponent strategy. Due to the time limit in the discussion period, we are not able to finish all comparisons between SES and EXP-STRATEGY.

As requested by reviewer SWZo, **we hereby promise to finish the following:**

2. **Complete experiments in FHP with more opponents (e.g., opponent $\Pr_{\rm shuffle}=0$).** (Half-done. Please refer to A2 in our second response to SWZo, and A1 in our third response to SWZo)
3. **Besides what we have explained in the responses to reviewer SWZo, we will discuss more in text about the differences and their significance. Our discussion will be based on new experiment results.** (Half-done. Please refer to A3 in our third response to SWZo)
4. **Add experiments on an additional game (such as Leduc)** (Done. Please refer to A2 in our third response to SWZo)

To be honest, it is hard to absolutely guarantee that a new theoretical result on EXP-STRATEGY can be achieved. For instance, the estimated opponent strategy is harder to analyze than just infoset distribution. But **we promise to**

1. **Try our best to theoretically characterize EXP-STRATEGY's guarantees in comparison to SES.**

Additionally, we also plan to complete experiments with various modeling errors $\epsilon$ to see how it impacts both algorithms, and which algorithm shows more robustness.

Thank all the reviewers again for their time and efforts on reviewing our work! If there are any additional questions or comments, we will be happy to have further discussions.

---

### Decision · Program_Chairs · 2022-01-20

**Decision:**

Reject

**Comment:**

The paper presents tackles the problem of finding strategies that are -- unlike Nash which is safe -- both safe (non-exploitable, to some extent) and able to exploit the opponent. The proposed solution is a convex combination of exploitation and safety that is efficient to compute. Overall, the paper is borderline. Given that the objective and its analysis are not especially surprising, a lot rides on the thoroughness of the empirical results, which could be improved.